# Potential of phenothiazines to synergistically block calmodulin and reactivate PP2A in cancer cells

**Ganesh Babu Manoharan**[ID][◔], **Sunday Okutachi**[◔], **Daniel Abankwa**[ID]*

Cancer Cell Biology and Drug Discovery Group, Department of Life Sciences and Medicine, University of Luxembourg, Esch-sur-Alzette, Luxembourg

◔ These authors contributed equally to this work.
* daniel.abankwa@uni.lu

**Data Availability Statement:** All relevant data are within the paper and its Supporting Information files.

**Funding:** The study was supported by internal financial and material funds of the University of

## Abstract

Phenothiazines (PTZ) were developed as inhibitors of monoamine neurotransmitter receptors, notably dopamine receptors. Because of this activity they have been used for decades as antipsychotic drugs. In addition, they possess significant anti-cancer properties and several attempts for their repurposing were made. However, their incompletely understood polypharmacology is challenging. Here we examined the potential of the PTZ fluphenazine (Flu) and its mustard derivative (Flu-M) to synergistically act on two cancer associated targets, calmodulin (CaM) and the tumor suppressor protein phosphatase 2A (PP2A). Both proteins are known to modulate the Ras- and MAPK-pathway, cell viability and features of cancer cell stemness. Consistently, we show that the combination of a CaM inhibitor and the PP2A activator DT-061 synergistically inhibited the 3D-spheroid formation of MDA-MB-231 (K-Ras-G13D), NCI-H358 (K-Ras-G12C) and A375 (B-raf-V600E) cancer cells, and increased apoptosis in MDA-MB-231. We reasoned that these activities remain combined in PTZ, which were the starting point for PP2A activator development, while several PTZ are known CaM inhibitors. We show that both Flu and Flu-M retained CaM inhibitory activity in vitro and in cells, with a higher potency of the mustard derivative in cells. In line with the CaM dependence of Ras plasma membrane organization, the mustard derivative potently reduced the functional membrane organization of oncogenic Ras, while DT-061 had a negligible effect. Like DT-061, both PTZ potently decreased c-MYC levels, a hallmark of PP2A activation. Benchmarking against the KRAS-G12C specific inhibitor AMG-510 in MIA PaCa-2 cells revealed a higher potency of Flu-M than combinations of DT-061 and a CaM inhibitor on MAPK-output and a strong effect on cell proliferation. While our study is limited, our results suggest that improved PTZ derivatives that retain both, their CaM inhibitory and PP2A activating properties, but have lost their neurological side-effects, may be interesting to pursue further as anti-cancer agents.

Luxembourg. All authors received salary from the University of Luxembourg. The funders had no role in study design, data collection and analysis, decision to publish, or preparation of the manuscript.

**Competing interests:** The authors have declared that no competing interests exist.

## Introduction

Several phenothiazines (PTZ) are approved as neuroleptics for the treatment of e.g. bipolar disorders. Structurally, they partially emulate dopamine, thus inhibiting dopamine D2 receptors, and affect also other monoamine neurotransmitter receptors [1]. In addition, PTZ are known to inhibit calmodulin (CaM), the ubiquitous calcium binding protein, which regulates the activity of hundreds of proteins [2–4]. CaM became a drug target in cancer, because of its profound involvement in regulating the cell cycle [3]. Thus, attempts were made in the late 1980s to exploit the CaM inhibiting activity of PTZ in cancer therapy development [5]. Such drug repurposing may offer a cheaper and faster access to live saving therapies. Since PTZ pass the blood brain barrier, the PTZ trifluoperazine was even tested in clinical trials to treat high grade gliomas [6].

Differential kinetic labelling experiments suggested that binding of trifluoperazine to $Ca^{2+}$/CaM perturbs lysines 75, 77 and 148, the same residues that are modified by the covalent CaM inhibitor ophiobolin A [7]. In agreement with the kinetic labelling studies, chemically reactive PTZ modified the same lysines [8]. Efforts were made to generate covalently binding derivatives of this compound class, as irreversible binding could significantly increase the potency of drugs [9]. For instance, the PTZ fluphenazine (Flu), was functionalized with a chlor-ethylamine, also known as mustard-group, to generate a covalently reacting fluphenazine-mustard (Flu-M) derivative [10]. Nitrogen mustards have been used for decades, because of their ability to crosslink the two strands of DNA [11]. The mustard-group is converted into a very reactive intermediate aziridinium ion at neutral or basic pH by intramolecular nucleophilic substitution of the leaving chloride [11]. The aziridinium then reacts with nearby nucleophiles, thus irreversibly linking the drug to the target, such as CaM.

While irreversible covalent inhibition can be regarded as the ultimate pharmacological inhibition strategy with potentially infinite ligand efficiencies, its exploitation in drug development was abandoned for many years. However, several successful drugs, such as aspirin and penicillin, rely on covalent inhibition mechanisms, and in recent years covalent inhibitor development has gained track again [9]. One of the most promising developments in this area are the first direct inhibitors of K-Ras-G12C, which are equipped with an acrylamide electrophile to specifically react with the cysteine in the mutant form of K-Ras. These compounds bind to a cryptic pocket under the SII loop that is only accessible in GDP-bound K-Ras [12]. Several realizations of such inhibitors exist from a number of companies, including the compounds ARS-1620, MRTX849 (Adagrasib) and AMG-510 (Sotorasib) [13–15]. While Adagrasib is still in clinical trials, Sotorasib has been performing well in the clinic and was recently granted accelerated approval by the FDA for the treatment of K-Ras-G12C mutant non-small cell lung cancers [16].

However, for other *KRAS* mutations and in other indications, there is still an urgent need for K-Ras inhibition strategies. Surrogate K-Ras targets, such as its trafficking chaperones CaM and PDE6D have emerged as interesting additional targets [17–19]. The interplay between K-Ras4B (hereafter K-Ras) and CaM is particularly relevant, as it was implicated in the ability of K-Ras to drive cancer cell stemness [20]. We previously showed that the natural product ophiobolin A (OphA), a covalent CaM inhibitor, can block stemness properties of cancer cells and K-Ras membrane organization [21]. We recently developed less toxic functional analogues of OphA, which confirmed this potential [22]. This lends an exciting new rationale to the targeting of CaM, which justifies renewed interest in CaM inhibitor development [23].

Additional MAPK-signaling targeting strategies may act complementary or even synergistically. The tumor suppressor protein phosphatase 2A (PP2A) inactivates several oncogenic

signaling pathways [24, 25]. Importantly, PP2A inactivation is required for the transformation of *RAS* mutant cancer cells, suggesting a particular importance of PP2A reactivation and Ras-pathway inhibition [26]. PP2A actually comprises over 80 different holoenzymes, which are each composed of a scaffolding A subunit, a regulatory B subunit and the catalytic C subunit [27]. The localization and specificity determining 15 distinct B subunits can be categorized into four families [28]. The activity of the C subunit is critically regulated by leucine carboxyl methyl transferase (LCMT), which reversibly attaches a methyl group to the C-terminal leucine to activate the phosphatase [29]. This is counteracted by phosphatase methylesterase 1 (PME-1), which demethylates this site [25]. Deregulation of the PP2A tumor suppressor in cancer typically occurs by the upregulation of endogenous inhibitors, such as cancerous inhibitor of PP2A (CIP2A), which selectively sequesters B56 family B subunits [30, 31]. Another endogenous inhibitor that is overexpressed, for instance in lung cancer, is SET [32].

Recently resistances against MEK-inhibition were shown to be overcome by novel PP2A activating drugs [33]. These phosphatase activators, promise to modulate the onco-kinome more broadly than kinase inhibitors [34]. The PP2A activators were derived from PTZ that were engineered for reduced central nervous system side-effects, such as sedation [35]. Mechanistically, the small-molecule activator of PP2A (SMAP) DT-061 was shown to bind to an inter-subunit region formed by all three PP2A subunits, thus specifically stabilizing PP2A-B56α heterotrimers [36]. The likewise PTZ derived iHAPs (improved heterocyclic activators of PP2A) function in the same way, however, slight differences in the B-subunit composition that is stabilized in the PP2A holoenzyme were found [37]. It is not known, however, whether these compounds can still bind to CaM and as such may act on multiple targets.

Given that both CaM and PP2A are pharmacological targets of PTZ, it is possible that these drugs could combine within one molecule a potential synergistic activity of CaM inhibitors acting on the K-Ras stemness signaling axis and PP2A-activators suppressing the Ras transforming activity [21, 33]. Here we examined whether CaM inhibition and PP2A activation could synergize and if this synergistic activity can be re-unified in PTZ-like compounds or covalently reacting derivatives thereof. We propose that the synergistic polypharmacology of PTZ that is suggested by our data may explain some of their profound anti-tumor effects [38, 39].

## Materials and methods

### Reagents

Sources of the compounds used in the study are given in parentheses, next to compounds name. Fluphenazine (Sigma, F0280000); Fluphenazine mustard (Enzo, BML-CA325-0050); Trifluoperazine (Sigma, 1686003); DT-061 (MedChem Express, HY-112929); Ophiobolin A (Santa Cruz, sc-202266); Calmidazolium (Santa Cruz, sc-201494); FTI-277 (BioVision, 2874); AMG-510 (MedChem Express, HY-114277); Trametinib (Bio-connect, SC-364639); Benzethonium chloride (Sigma-Aldrich, 53751). DMSO was from PanReac-AppliChem (cat. no. A3672, ITW Reagents). Caspase-Glo 3/7 assay kit was from Promega (G8090).

### ATARiS gene dependence score

ATARiS gene dependency scores for each gene of interest were extracted from the publicly available database of the project DRIVE (https://oncologynibr.shinyapps.io/drive/) [40]. The normalized viability data for the siRNA knockdown of each gene of interest were downloaded from the project DRIVE and a double gradient heatmap plot was generated using Prism (GraphPad).

## 3D spheroid assays

3D spheroid formation assays were performed in 96-well low-attachment, suspension culture plates (cat. no. 655185, Cellstar, Greiner Bio-One) under serum free conditions. About 1000 cells per well were seeded in 50 μL RPMI medium (cat. no. 52400–025, Gibco, ThermoFisher Scientific) containing 0.5% (v/v) MethoCult (cat. no. SF H4636, Stemcell technologies), 1X B27 (cat. no. 17504044, Gibco, ThermoFisher Scientific), 25 ng/mL EGF (cat. no. E9644, Sigma-Aldrich) and 25 ng/ mL FGF (cat. no. RP-8628, ThermoFisher Scientific). Cells were cultured for 3 days and then treated with compounds or vehicle control (DMSO 0.1% (v/v) in growth medium) for another 3 days. The cells were supplemented with fresh growth medium on the third day together with the drug treatment. Spheroid formation efficiency was analyzed by alamarBlue assay reagent (cat. no. DAL1100, ThermoFisher Scientific).

A 10% (v/v) final volume of alamarBlue reagent was added to each well of the plate and incubated for 4 h at 37˚C. Then the fluorescence intensity ($\lambda_{excitation}$ 560 ± 5 nm and $\lambda_{emission}$ 590 ± 5 nm) was measured using the FLUOstar OPTIMA plate reader (BMG Labtech, Germany). The obtained fluorescence intensity data were normalized to vehicle control corresponding to 100% sphere formation and the signal after 100 μM benzethonium chloride, which killed all cells (i.e. maximum inhibition of sphere formation).

## Bliss synergism experiments

To assess the synergistic potential of our compounds, we performed a full dose response analysis of one inhibitor (calmidazolium) and maintained a fixed concentration of the other inhibitor (DT-061) at either 1 or 2 μM. The drug response profiles obtained for the combinations were then compared against the profiles of each single agent using the SynergyFinder platform [41]. SynergyFinder (https://synergyfinder.fimm.fi) is a stand-alone web-application for interactive analysis and visualization of drug combination screening data [41]. We employed the Bliss model [42], which generates the multiplicative effects of single agents as if they acted independently in scoring our pairwise combinations

$$S_{BLISS} = E_{A,B,...,N} - (E_A + E_B + \ldots E_N - E_A E_b - E_A E_N - E_B E_N - \ldots - E_A E_B \ldots E_N).$$

Here, $E_A, E_B, \ldots, E_N$ are the measured responses of the single drugs, while *b* is the doses of the single drugs required to produce the combination effect $E_{A,B,\ldots,N}$.

## Drug sensitivity score (DSS) analysis

The Drug Sensitivity Score (DSS) is a more robust parameter than $IC_{50}$ or $EC_{50}$ and measures essentially the normalized area under the curve of dose-response data [43]. The normalized % inhibition data of the BRET assay or the raw intensity data of the 2D-monolayer assay were uploaded to the DSS pipeline website, Breeze (https://breeze.fimm.fi/) [22, 44]. The output file from the Breeze platform containing $DSS_3$ and several other drug sensitivity measures including $IC_{50}$ or $EC_{50}$ and AUC was downloaded after the simulation.

## 2D cell viability assays

HEK-293 EBNA and MIA PaCa-2 cells were cultured in DMEM and MDA-MB-231 cells in RPMI medium supplemented with 10% (v/v) FBS (cat. no. 10270–098, Gibco, ThermoFisher Scientific), 2 mM L-glutamine (cat. no. 25030–024, ThermoFisher Scientific), 1% penicillin-streptomycin (cat. no. 15140122, Gibco, ThermoFisher Scientific). Cells were plated into 96-well F-bottom cell culture plates (cat. no. 655180, Cellstar, Greiner Bio-One) at a density of 1000 cells per well and grown for 24 h. Compounds at indicated concentration or as a control

DMSO at 0.2% (v/v) in growth medium were applied for 72 h then the cell viability was measured using alamarBlue assay. A 10% (v/v) final volume of alamarBlue reagent was added to each well of the plate and incubated for 4 h at 37°C. Then the fluorescence intensity ($\lambda_{excitation}$ 560 ± 5 nm and $\lambda_{emission}$ 590 ± 5 nm) was measured using the FLUOstar OPTIMA plate reader (BMG Labtech). The obtained fluorescence intensity data were normalized to vehicle control (100% viability).

## Flow cytometry-based apoptosis assay

MDA-MB-231 cells were grown for 24 h in complete RPMI medium. The next day the cells were treated with inhibitors for 24 h. Then the cells were collected, washed, and suspended in ice-cold PBS (cat. no. 14190–094, Gibco, ThermoFisher Scientific). The cells were labelled using the Annexin V-FITC / 7-AAD apoptosis-detection kit (Beckman-Coulter, cat. no. B60224-AB) according to the manufacturer's instruction. The labelled cells were then transferred to a clear 96-well plate (cat. no. 655180, Greiner) and analyzed using Guava easyCyte (Luminex) cytometer.

## Fluorescence polarization assay

The fluorescence polarization assay was performed as previously described [22, 23]. A complex of recombinant bovine CaM (cat. no. 208690, Merck), with an amino acid sequence identical to the human isoform, and fluorescein-labelled PMCA peptide was added to a 3-fold dilution series of inhibitor in an assay buffer (20 mM Tris Cl pH 7.5, 50 mM NaCl, 1 mM $CaCl_2$ and 0.005% (v/v) Tween 20) in a black low volume round bottom 384-well plate (cat. no. 4514, Corning). After 24 h incubation, the fluorescence anisotropy ($\lambda_{excitation}$ 482 ± 8 nm and $\lambda_{emission}$ 530 ± 20 nm) was recorded in a Clariostar (BMG labtech) plate reader. The $IC_{50}$ value was derived by fitting the log concentration of inhibitor vs fluorescence anisotropy signal in Prism (GraphPad) software. The $IC_{50}$ value was converted into $K_d$ as described previously [45].

## BRET assays

The detailed BRET methodology and plasmids encoding Rluc8-K-RasG12V, Rluc8-H-RasG12V, GFP2-K-RasG12V, GFP2-H-RasG12V and GFP2-CaM were recently described [22]. About 100,000 to 150,000 HEK-293 EBNA [46] cells were seeded per well of a 12-well plate in 1 mL of DMEM containing 10% (v/v) FBS, 0.5 mM L-glutamine and 1% (v/v) penicillin-streptomycin and grown for overnight. Next day 1 µg of BRET sensor plasmids were transfected using 3 µL of jetPRIME reagent (cat. no. 114–75, Polyplus). The ratio of acceptor to donor plasmid (A/D plasmid ratio) for each experiment is indicated in the figure legends. After 24 h of transfection, cells were treated with vehicle control (DMSO 0.2% (v/v) in growth medium) or compounds at the specified concentration for 24 h. Then the cells were collected, washed and re-plated in PBS into flat bottom, white 96-well plates (cat. no. 236108, Nunc, ThermoFisher Scientific). First the fluorescence intensity ($\lambda_{excitation}$ 405 ± 10 nm and $\lambda_{emission}$ 515 ± 10 nm) of GFP2 was measured as it is directly proportional to the acceptor concentration. Next the BRET readings were taken in well-mode by adding coelenterazine 400a (cat. no. C-320, GoldBio) to a final concentration of 10 µM and luminescence emission intensities were simultaneously recorded at 410 ± 40 nm and at 515 ± 15 nm. The raw BRET ratio was calculated as the BRET signal measured at 515 nm divided by the emission signal measured at 410 nm. The BRET ratio was obtained by subtracting the raw BRET ratio from the background BRET signal measured for cells expressing only the donor.

## Western blotting

MDA-MB-231 cells were grown for 24 h in RPMI supplemented with 10% FBS treated with inhibitors for 24 h. Then the cells were lysed using RIPA buffer (50 mM Tris, 150 mM NaCl, 0.1% (v/v) SDS, 1% (v/v) Triton X-100, 1% (v/v) NP40) supplemented with protease inhibitor cocktail (ThermoScientific), PhosStop (Roche) and 2 mM DTT. MIA PaCa-2 cells were grown in DMEM supplemented with 10% FBS for 24 h. Next day the cells were treated with inhibitors for 2 h in a serum free medium and then stimulated for 10 min with EGF (50 ng/ mL) and then lysed.

The cell lysates were boiled with Laemlli buffer for 5 min at 95˚C and resolved on Mini-PROTEAN precast gels (Bio-Rad). For analyzing ERK phosphorylation, cell lysates were resolved on a 12% homemade acrylamide gel. Proteins were subsequently transferred onto a nitrocellulose membrane (Bio-Rad) using the Trans-Blot Turbo Transfer system (Bio-Rad) and probed with a primary antibody. The primary antibodies employed were anti-c-MYC (Cell Signaling, #5605 at 1:1000), anti-PARP (Cell signaling, #9542 at 1:1000), anti-vinculin (Cell Signaling, #13904, at 1:1000), anti-GAPDH (Sigma, G8795, at 1:10,000), anti-ERK (Cell Signaling, #9102 at 1:1000) and anti-pERK(Tyr204) (Santa Cruz, sc-7383 at 1:400). An anti-mouse or anti-rabbit IRDye 800CW or 680RD secondary antibody (LI-COR) was used subsequently to develop the membrane and the proteins were detected using an Odyssey CLx system (LI-COR). The level of proteins was densitometrically quantified from images of membranes analyzed using ImageJ or ImageLab (Bio-Rad) software.

## Data analysis

All data analysis was performed using Prism (GraphPad) version 9. The number of independent biological repeats, n, for each data set is provided in the relevant figure legend. Unless otherwise stated, statistical significance was evaluated using one-way ANOVA. A p-value of $< 0.05$ is considered statistically significant and the statistical significance levels are annotated as: * = $p < 0.05$; ** = $p < 0.01$; *** = $p < 0.001$; **** = $p < 0.0001$, or ns = not significant.

## Results

### Phenotypic assessment of CaM inhibition and PP2A activation reveal synergistic drug interaction commensurate with phenothiazine effects

We hypothesized that PTZ could synergistically combine the activities of CaM inhibition and PP2A activation. To assess this possibility, we first analyzed the combinatorial effect of the potent non-covalent inhibitor calmidazolium (CMZ) and the PP2A activating SMAP, DT-061 (Scheme 1), on the growth of *KRAS* mutant MDA-MB-231 breast and NCI-H358 lung cancer cell 3D spheroids, as well as *BRAF* mutant A375 skin cancer cell spheroids. These cell lines showed a spectrum of genetic dependencies on *KRAS*, *HRAS*, *BRAF*, *CALM1-3* (the three human CaM genes), the alpha isoform of the C subunit of the PP2A enzyme (*PPP2CA*) and an endogenous inhibitor of PP2A in cancer, *SET* (Fig 1A). Of note, only the MDA-MB-231 breast cancer cell line showed a dependency on *KRAS*, *CALM3* as well as *PPP2CA*.

While both CMZ or DT-061 alone showed low micromolar inhibition in all cell lines, their combination significantly lowered the average IC$_{50}$ across all tested cell lines (Figs 1B and S1A–S1C). Scoring for synergistic activity of the combination suggested high synergism of these combinations in all three cell lines (Figs 1C and S1D). Importantly, a similar level of inhibition as with the combination was achieved using only the PTZ trifluoperazine (Figs 1B and S1A–S1C). Interestingly, both the combination and the single agent trifluoperazine, were as potent as the K-Ras-G12C specific covalent inhibitor ARS-1620 in NCI-H358 cells.

**Scheme 1. Structure of compounds employed in this study.** Shown are the calmodulin inhibitors calmidazolium (used as chloride salt and shown without counterion) and W7, the PP2A activator DT-061, and the phenothiazines trifluoperazine, fluphenazine and fluphenazine mustard. Note the relatedness between DT-061 and phenothiazines, which all feature the tricyclic moiety characteristic of phenothiazines.

Next, the effect of the compounds on apoptosis was assessed. Flow cytometric analysis using annexin V-FITC and 7-AAD (DNA) staining of MDA-MB-231 cells, revealed that the combination of low dose of DT-061 and CMZ significantly increased the apoptotic cell population to 50% compared to the single agent treatment of DT-061 (12%) (**Figs 1D and S2A–S2F**). Similarly, the combination of DT-061 and CMZ significantly increased the level of caspase-3/7 up to 2.5-fold compared to single agent treatment with DT-061 or CMZ alone (**Fig 1E**) [33]. In support of the higher potency of the combination, as compared to the single agents, cleavage of poly (ADP-ribose) polymerase (PARP), a hallmark of apoptosis, was higher for the combination than for DT-061 or CMZ alone (**Figs 1F and S2G**) [47].

These data, therefore, tentatively suggest that it could be beneficial to employ PTZ derivatives with dual action on CaM and PP2A against cancer cells.

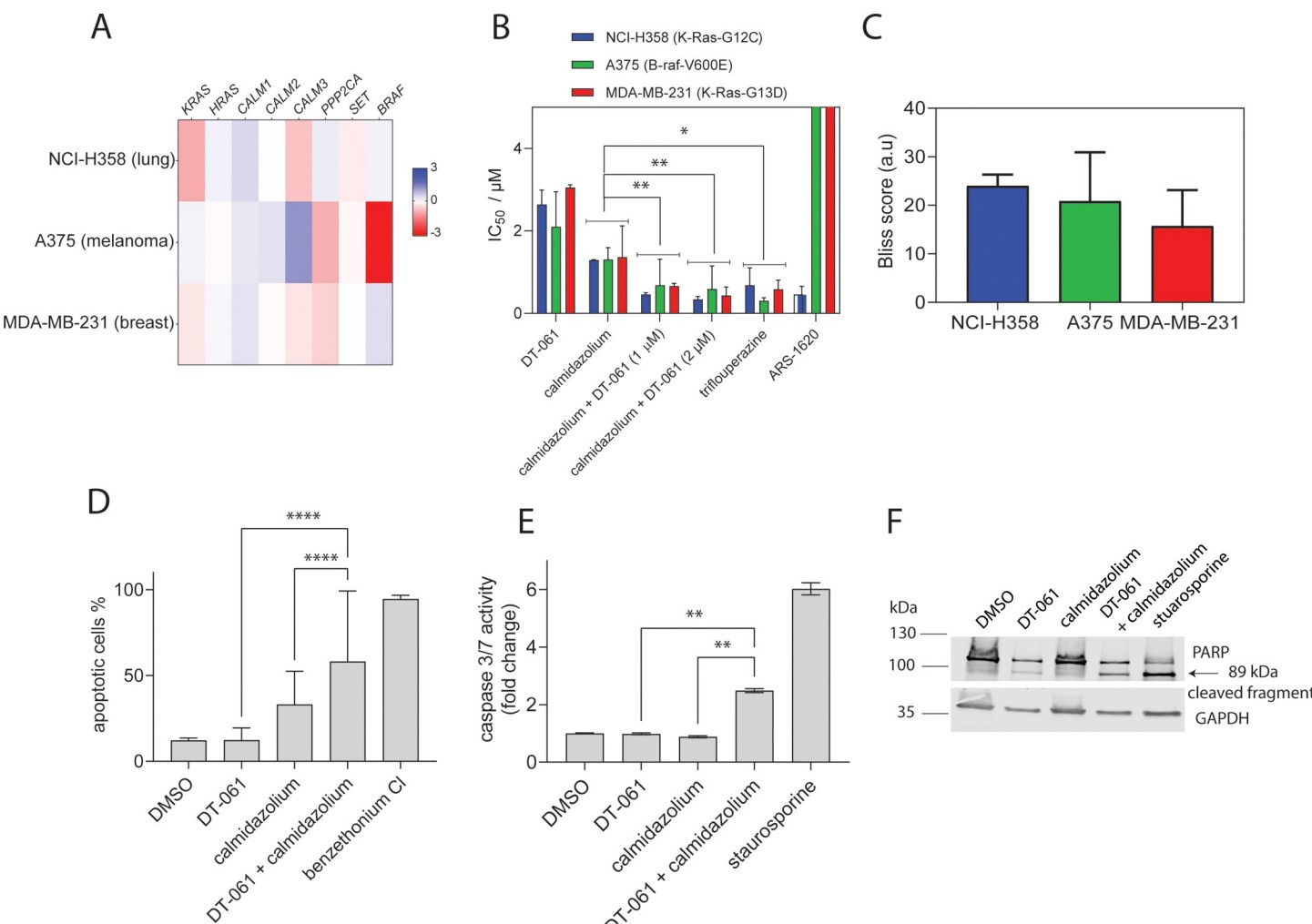

**Fig 1. Cellular assessment of CaM inhibitors and PP2A activators suggests their synergistic potential against Ras pathway dependent cancer cell lines. (A)** Heatmap of ATARiS gene sensitivity scores of two *KRAS* dependent cell lines (NCI-H358 and MDA-MB-231) and a *BRAF* dependent cell line (A375). Negative values (shaded red) indicate sensitivity of the cell line proliferation to the knockdown of shown genes, whereas positive scores (shaded blue) indicate insensitivity. **(B)** IC_{50} values for indicated compounds in Ras pathway mutant cell lines NCI-H358, MDA-MB231 and A375 grown as 3D spheroids under serum free conditions. Compounds were tested either as single agent at a concentration range of 0.2 μM– 10 μM (calmidazolium), 0.6 μM– 40 μM (DT-061), 0.2 μM– 40 μM (trifluoperazine) and 0.6 μM– 40 μM (ARS-1620) or in combination, applying the whole concentration range of calmidazolium combined with 1 μM or 2 μM of DT-061, as indicated. Data represent mean values ± SD, n = 2. Statistical comparisons were done using the average values across all three cell lines. **(C)** Bliss synergism scores for combinatorial effects of calmidazolium and DT-061 in KRAS- or BRAF-mutant cancer cell lines. Data represent mean values ± SD, n = 2. Positive scores indicate synergistic drug interactions whereas negative scores would denote antagonistic drug interactions. A score of zero would indicate no antagonistic or synergistic effect. **(D)** Flow cytometric analysis of apoptosis in MDA-MB-231 cells. MDA-MB-231 cells were treated with DT-061 (10 μM) or calmidazolium (5 μM) or their combination for 24 h and then labelled with Annexin V-FITC and 7AAD. Benzethonium chloride and DMSO treated cells were used as positive and negative controls, respectively, for setting the gates. Percentage of total apoptotic cells (early and late stage) from two biological repeats (mean ± SD) is presented. Statistical analysis was performed using Fisher's exact test comparing the total number of viable and apoptotic cells from two biological repeats. **(E)** Effect of inhibitors on cellular apoptosis studied using Caspase-3/7Glo assay. MDA-MB-231 cells were treated with DT-061 (5 μM), calmidazolium (5 μM), combination of DT-061 and calmidazolium (5 μM each), and staurosporine (1 μM) for 24 h. The caspase-3/7 activity was measured as a luminescence read-out and normalized to the vehicle control, 0.05% (v/v) DMSO. Data represent mean values ± SD, n = 2. **(F)** Western blot analysis of PARP-cleavage activity in MDA-MB-231 cells treated with DMSO (0.2% (v/v)), DT-061 (20 μM), calmidazolium (5 μM), combinations of DT-061 (10 μM) and calmidazolium (5 μM), and staurosporine (1 μM) for 24 h. GAPDH detection was used as loading control.

## A covalent phenothiazine derivative binds to CaM and shows enhanced disruption of the cellular K-RasG12V/ CaM interaction

Targeted covalent inhibitors are attracting again more attention as an interesting option in drug development [9]. We therefore tested whether a covalent derivative of the PTZ

fluphenazine (Flu), with addition of an electrophilic mustard group, fluphenazine mustard (Flu-M), can increase the efficacy of its parental compound.

In order to measure direct inhibition of CaM, we used a fluorescence polarization assay where displacement of a fluorescein-labelled CaM binding peptide, which was derived from plasma membrane $Ca^{2+}$/ATPase (PMCA), by CaM inhibitors is measured [23]. Both parental, non-covalently reacting Flu and covalently reacting Flu-M exhibit submicromolar affinities to CaM (**Fig 2A** and **Table 1**). By contrast, essentially no binding to CaM was detected for the SMAP DT-061, demonstrating that it does not target CaM.

CaM can act as a trafficking chaperone of K-Ras by shielding the hydrophobic farnesyl tail from the aqueous environment of the cytoplasm, thus effectively increasing the diffusibility of K-Ras [22, 48]. In order to detect an effect of the CaM inhibitors on the K-Ras/ CaM interaction in cells, we established a BRET assay [22], where we genetically fused Rluc8 to the N-terminus of K-RasG12V and GFP2 to the N-terminus of CaM. Both the covalently reacting CaM inhibitor OphA and the non-covalent, highly potent CMZ reduced the BRET signal when tested across a wider concentration range, in agreement with their CaM inhibiting activity (**Fig 2B**). In order to robustly distinguish the magnitude of these effects, we analyzed the dose response data using a normalized area under the curve measure, the $DSS_{3\ BRET}$-score [22, 43]. These data revealed that addition of the electrophile in Flu-M significantly increased the intracellular activity against CaM/ K-RasG12V as compared to its non-covalent counterpart Flu by more than a factor of two (**Fig 2C**). Yet the potency remained below that of OphA, which however has itself significant off-target activities that may be associated with its broad toxicity spectrum [22].

## Cellular BRET assays confirm Ras membrane organization disrupting properties of fluphenazine mustard

By genetically fusing BRET-donor signal enabling Rluc8 and the acceptor GFP2 to Ras proteins, we established BRET-assays, which can measure the loss of the functional membrane organization of Ras after inhibition of its trafficking chaperone CaM [22]. Tight packing of BRET-luminophore labelled RasG12V in plasma membrane nanoclusters leads to high BRET-levels, similar to what we previously showed [49, 50]. Loss of nanoclustering, or any process upstream, including proper plasma membrane trafficking or lipid modification of Ras, reduces this nanoclustering-dependent BRET signal. Consistently, treatment with the farnesyl-

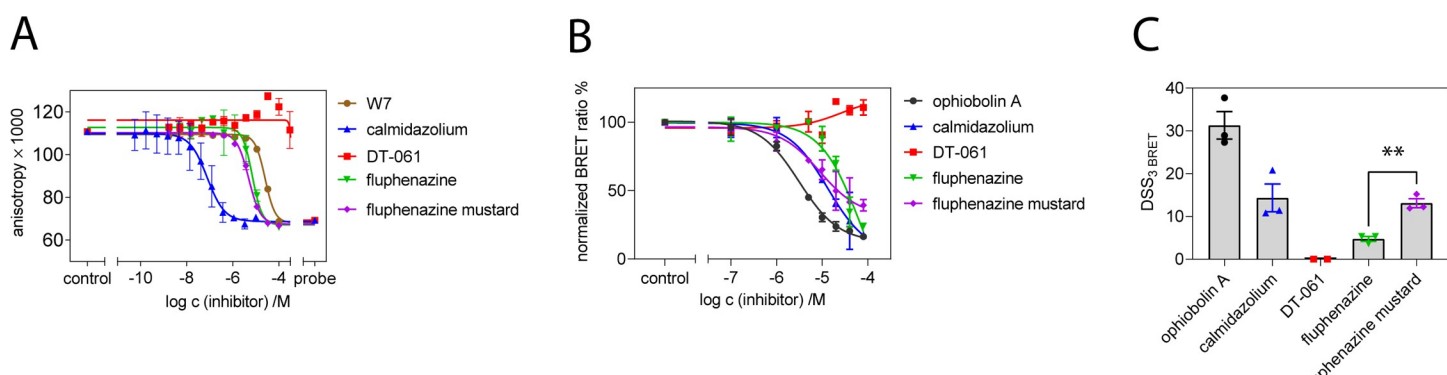

**Fig 2. Assessment of the in vitro inhibition of CaM and in cellulo disruption of K-Ras/ CaM binding. (A)** CaM inhibition was assessed using 100 nM CaM and displacement of 10 nM F-PMCA CaM-binding peptide by inhibitors using fluorescence polarization measurements. Data represent mean values ± SD, n = 2. **(B)** Dose-response analysis of indicated inhibitors at 0.1–80 μM using the Rluc8-K-RasG12V/ GFP2-CaM BRET assay in HEK-293 EBNA cells. The acceptor/ donor plasmid ratio was 9/1. Data represent mean values ± SD, n ≥ 2. **(C)** $DSS_{3\ BRET}$ i.e. normalized area 'under' the inhibition curve values (hence, here actually above the curve) of dose response data shown in (B). Statistical analysis was performed using unpaired t-test.

**Table 1. CaM-binding affinity of compounds from fluorescence anisotropy data in Fig 2A (mean ± SD, n = 2).**

| Compound | $IC_{50}$ /μM | $K_d$ |
|---|---|---|
| W7 | 25 ± 2 | 1.5 ± 0.1 μM |
| fluphenazine | 7 ± 2 | 0.4 ± 0.1 μM |
| fluphenazine mustard | 4.9 ± 0.5 | 0.29 ± 0.03 μM |
| DT-061 | > 3700 | > 200 μM |
| calmidazolium | 0.08 ± 0.04 | 1.1 ± 0.1 nM |

transferase inhibitor that is known to selectively affect H-Ras more than K-Ras [49], reduced the nanoclustering-dependent BRET signal of H-RasG12V more than that of K-RasG12V (**Fig 3A and 3B**). In agreement, with our previous results [21, 22], CaM inhibitors CMZ and OphA showed an inverse albeit less pronounced selectivity, reducing the K-RasG12V BRET signal more than that of H-RasG12V (**Fig 3A and 3B**). This was confirmed across a broader concentration range, using the $DSS_{3\ BRET}$-score analysis [22] (**Figs 3C and S3**).

Neither the PP2A activator DT-061, nor Flu had an effect on the RasG12V-BRET signals (**Figs 3 and S3**). By contrast, treatment with the covalently-reacting derivative Flu-M led to an even stronger effect than CMZ (**Fig 3**), but against both K-RasG12V and H-RasG12V biosensors.

These data suggest that addition of the mustard group can significantly increase the potency of Flu-M against Ras membrane organization in cells.

## Fluphenazine mustard decreases c-MYC expression, ERK-signaling and cell proliferation

PP2A enhances the ubiquitin-mediated degradation of c-MYC [51], and PP2A activating SMAPs, such as DT-061, decrease c-MYC expression levels [33, 52]. We therefore assessed the

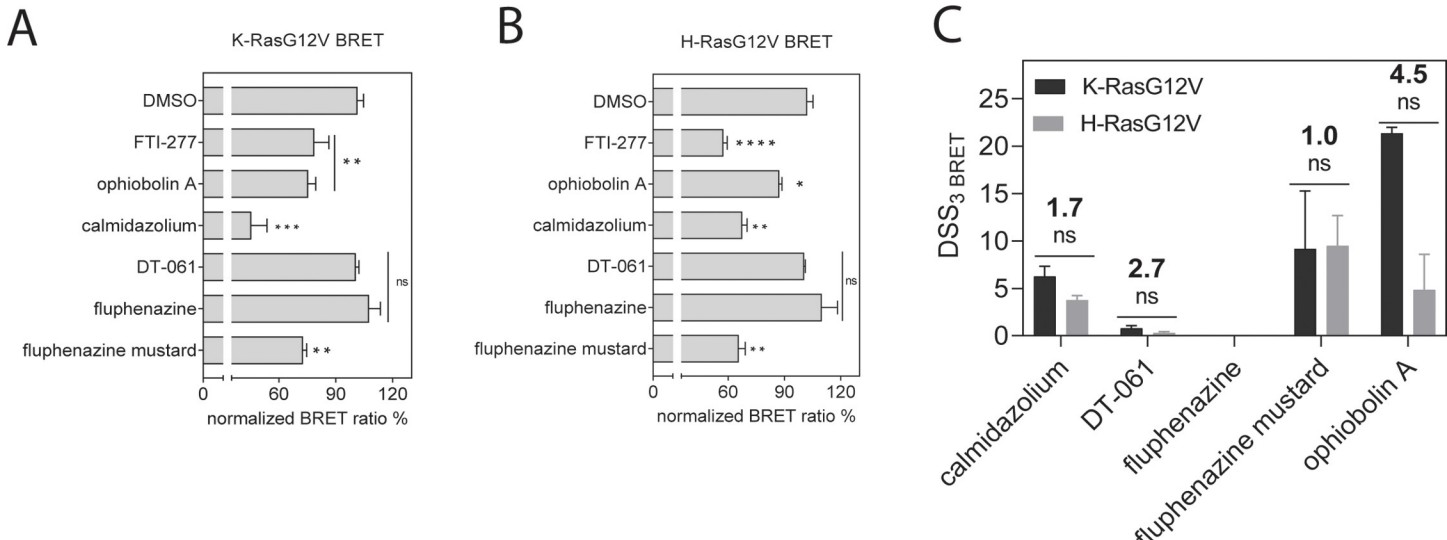

**Fig 3. BRET assay reveals high potency of fluphenazine mustard against Ras-membrane organization in cells.** (**A**) Testing of indicated compounds at 20 μM for 24 h exposure in KRasG12V (**A**) and H-RasG12V (**B**) nanoclustering-BRET assays. Controls are FTI-277 (1 μM), OphA (2.5 μM), calmidazolium (20 μM) and DT-061 (20 μM). The acceptor/donor plasmid ratio of GFP2- and Rluc8-tagged RasG12V was 4/1. Data represent mean values ± SD, n ≥ 2. (**C**) $DSS_{3\ BRET}$ values derived from dose response data for calmidazolium, DT-061, Flu, Flu-M (0.1 μM– 80 μM) and OphA (0.3 μM– 20 μM) using K-RasG12V and H-RasG12V nanoclustering-BRET assays. The acceptor/donor plasmid ratio was 4/1. Data represent mean values ± SD, n ≥ 2.

c-MYC levels after treatment with Flu and Flu-M. Both Flu and Flu-M decreased the c-MYC level in MDA-MB-231 cells (**Figs 4A and 4B** and **S4**) comparable to that of DT-061, supporting their PP2A activation.

Next, we compared the ERK signaling activity of Flu-M with AMG-510, a clinically employed KRAS-G12C inhibitor, in MIA-PaCa2 cells. In agreement with the Ras membrane organization disrupting activity (**Fig 3**), Flu-M inhibited pERK levels significantly more potently than CMZ and DT-061, alone or in combination (**Figs 4C and 4D** and **S5**). Hence Flu-M appears to also combine the synergistic activity of the CaM inhibitor and PP2A activator in this context. For comparison, MAPK-signaling inhibition by the clinically employed KRAS-G12C inhibitor AMG-510 or MEK inhibitor trametinib was assessed (**Figs 4C and 4D** and **S5**). Both of these inhibitors were at least 20-times more potent than Flu-M in this assay.

However, when tested for its effect on cell proliferation, both Flu and Flu-M appeared relatively potent as compared to AMG-510, with Flu-M being significantly more potent against MIA PaCa-2 cell proliferation than Flu (**Figs 4E** and **S6**). Given that Flu-M has a moderate inhibitory activity on pERK-levels, the relatively potent inhibition of cell proliferation is in agreement with multiple targeting mechanisms of this compound.

## Discussion

Here we provided evidence that PTZ can integrate the synergistic cancer cell inhibitory activities of CaM inhibitors and PP2A activators within one compound. This activity can

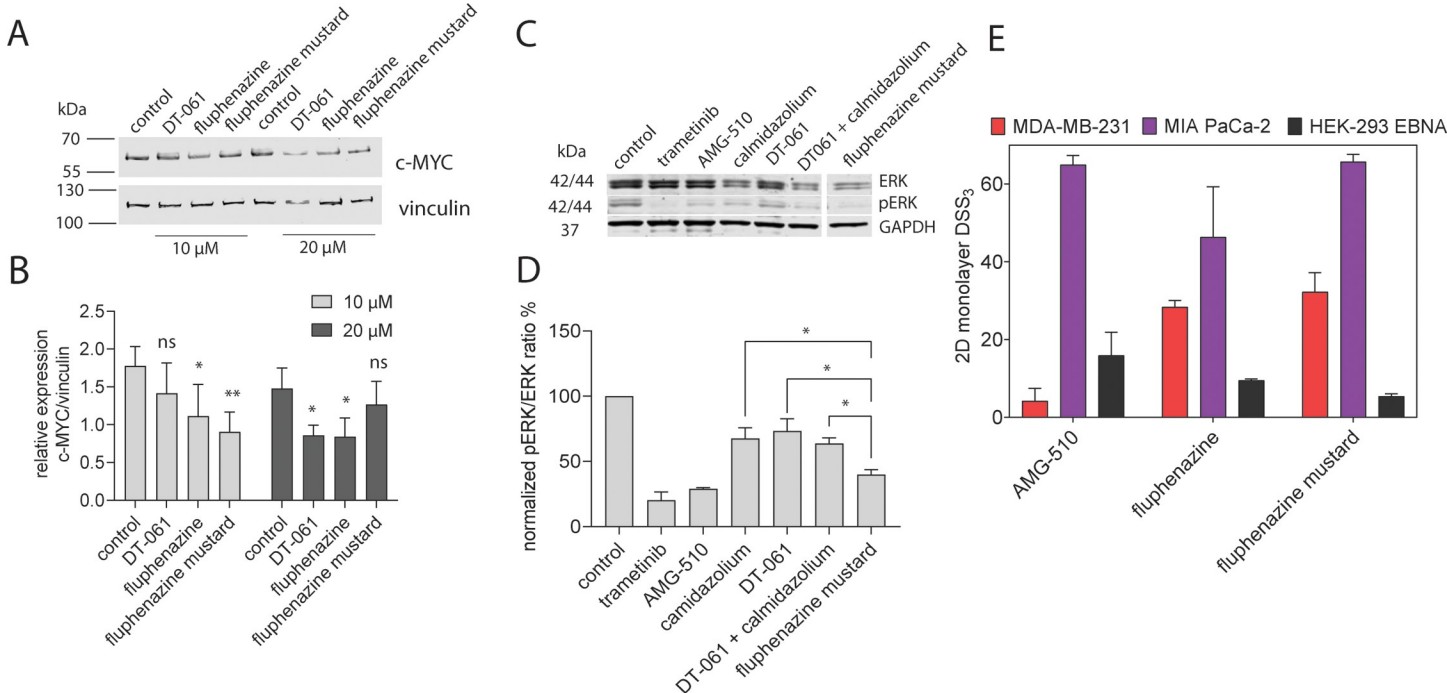

**Fig 4. PP2A activity, MAPK signaling and anti-proliferative effects of Flu-M. (A)** Expression levels of c-MYC was measured in MDA-MB-231 after treatment with 10 μM or 20 μM of inhibitors for 24 h. Vinculin was used as loading control. A representative blot from 4 biological repeats is presented. **(B)** Expression of c-MYC relative to vinculin levels was analyzed. 0.1% (v/v) and 0.2% (v/v) of DMSO was used as control for 10 μM and 20 μM compound treatments, respectively. Data represent mean ± SD of 4 biological repeats. Statistical analysis was performed using unpaired t-test. **(C)** MAPK signaling output was measured in MIA PaCa-2 cells after 2 h treatment of trametinib (1 μM), AMG-510 (1 μM), calmidazolium (20 μM), DT-061 (20 μM), combination of DT-061 and calmidazolium (each 10 μM) and fluphenazine mustard (20 μM) in the absence of serum followed by EGF stimulation (50 ng/ml) for 10 min. 0.2% (v/v) DMSO was used as vehicle control. Representative blot from three independent biological repeats. **(D)** Densitometric quantification of the pERK/ERK ratio of all three biological repeats (mean values ± SD) normalized to control. Statistical analysis was performed using unpaired t-test. **(E)** $DSS_3$ measuring the anti-proliferative effects of AMG-510 (0.003–40 μM), fluphenazine (0.6–80 μM) and fluphenazine mustard (0.6–80 μM). Results represent mean values ± SD, n = 3.

be potentiated using a PTZ, such as Flu-M, equipped with an electrophile that can covalently inactivate one or both of these targets. At this point, it is not clear, whether the covalent derivative Flu-M also covalently engages PP2A. CaM inhibition by PTZ is supported by in vitro and in cellulo data. By contrast, the SMAP DT-061 does not inhibit CaM. Given that CaM is a trafficking chaperone of K-Ras, CaM inhibition disrupts functional Ras-membrane organization and downstream MAPK-output. This is supported by the relatively high potency of Flu-M against 2D proliferation of MIA PaCa-2 cells, while it is also consistent with the engagement of several targets, such as PP2A. One clear limitation of our study is that we have not directly examined PP2A activation by PTZ, for example by reconstitution of the holoenzyme, but instead employed surrogate measures such as c-MYC degradation.

Importantly, addition of the electrophile did not appear to increase unspecific toxicity in cells (**Fig 4E**). Taken together with the more potent inhibition of strongly KRAS-dependent MIA PaCa-2 cells and the increased CaM inhibition in cells (**Fig 2C**), a significant contribution of CaM inhibition in the cell killing phenotype of Flu-M is apparent. Furthermore, the dual activity of Flu-M against K-RasG12V and H-RasG12V membrane anchorage BRET may explain the high potency (**Fig 3C**). It is currently not clear, why also the H-RasG12V BRET-signal was affected, while we previously noted a clear K-Ras selectivity of non-covalent and covalent CaM inhibitors [22]. It may suggest that other targets are engaged by this compound that affect membrane organization of both Ras isoforms.

While our study is quite limited, it suggests that preservation of the CaM inhibitory activity of PTZ derivatives that can also activate PP2A is desirable. Our data show however, that DT-061 does not bind to CaM and would therefore have lost the synergistic potential. We propose that the preservation of the here investigated polypharmacology of PTZ could allow for a higher, yet selective cancer cell killing activity.

This is supported by substantial evidence for the anti-cancer activity of PTZ. Several studies have shown that PTZs inhibit the growth of various cancer types including glioblastomas, melanoma, colorectal cancer and acute lymphoblastic leukemia [38, 53, 54]. The ability of PTZs to effect multiple biological consequences, such as the disruption of membrane functions, DNA repair, cell cycle regulation, apoptosis and multiple signaling pathways, indicates the potential of this class of compounds as viable starting points for novel anti-cancer agents [55, 56]. Interestingly, the dopamine receptor mRNA and protein expression levels are elevated in a variety of cancers including cervical, esophageal, glioma, breast cancer and lung cancer [57–61]. Therefore, also this target may play a role in the anti-cancer effect of PTZ.

We could show that the mustard derivative of the PTZ fluphenazine, Flu-M, which has a quite reactive electrophile, affords a comparable cancer cell selectivity as its non-covalent counterpart. It can be expected that further optimizations here, could allow for improved chemical selectivity, for instance by using a softer electrophile such as an acrylamide. One particular advantage of covalent inhibitors is that the covalent bonding step might actually improve selectivity for a desired target, as only that target may exhibit proper nucleophile constellation near the non-covalent initial binding site.

Alternative PP2A activators have been identified, such as the sphingosine 1-phosphate receptor modulator FTY720 (Fingolimod), which is FDA approved for multiple-sclerosis. It binds to the endogenous PP2A inhibitor SET, thus efficaciously reducing leukemic burden in mice [62]. This potential has been recapitulated by an improved derivative, CM-1231, which shows no more cardiotoxicity as compared to Fingolimod and could be developed for treatment of leukemic patients with SET overexpression [63]. It remains to be seen whether Fingolimod derivatives or PTZ derivatives can be developed into new and improved PP2A activators.

## Conclusion

Our data suggest rethinking of the design and target spectrum of PTZ-derived SMAPs and related PP2A activators. We propose that it is beneficial to endow PTZ derivatives with both CaM inhibitory and PP2A activating properties in one molecule. Starting from existing PTZ, modifications that eliminate the neurological side-effects could be an important and possibly sufficient first step. Thus, it may be possible to generate PTZ derivatives with increased and selective cancer cell killing activity.

## Supporting information

**S1 Fig. Data for synergism analysis.** (**A-C**) Dose response curves for indicated inhibitors in Ras pathway mutant cell lines NCI-H358 (A), A375 (B) and MDA-MB-231 (C). Compounds were tested as either single agents at concentration ranges of 0.2 μM– 10 μM (calmidazolium), 0.6 μM– 40 μM (DT-061), 0.2 μM– 40 μM (trifluoperazine) or 0.6 μM– 40 μM (ARS-1620), or in combination at a full dose response range of 0.2 μM– 10 μM for calmidazolium with 1 or 2 μM of DT-061 added to all test conditions. Data represent mean values ± SD, n = 2. The data were fit into log (inhibitor) vs variable response (four parameters) function in the Prism (GraphPad) to obtain the dose response curves. The actual curve fitting for Bliss calculation was done on the SynergyFinder website (https://synergyfinder.fimm.fi/). (**D**) Representative Bliss synergism heatmaps for combinatorial effects of calmidazolium and DT-061 in Ras mutant cancer cell lines. Heatmaps with positive scores (shaded red) indicate synergistic drug interaction whereas, heatmaps with negative scores (shaded green) indicates antagonistic drug interaction.
(TIF)

**S2 Fig. Flow cytometric analysis of apoptosis and Western blot analysis of PARP in MDA-MB-231 cells. (A)** Percentage of viable cells (non-stained), early apoptotic cells (Annexin V-FITC stained), late apoptotic cells (both annexin V-FITC and 7AAD stained) and necrotic and damaged cells (7AAD stained), representing each quadrant of a dot plot is presented. Data represent mean ± SD of two independent biological repeats. **(B-F)** Representative dot plots of DMSO (B), benzethonium chloride (C), DT-061 (D), calmidazolium (E), and combination of DT-061 and calmidazolium (F) from one biological repeat. (**G**) Uncropped Western blot membrane presented in **Fig 1F**. After the transfer the membrane was cut between 70 and 55 kDa and the top and bottom blots were probed with anti-PARP and anti-GAPDH antibody, respectively. X and marker indicate non-related sample and PageRuler (ThermoScientific) protein ladder, respectively.
(TIF)

**S3 Fig. K-Ras and H-Ras nanoclustering BRET assay data of Fig 3C.** (**A, B**) Dose response curves of fluphenazine and fluphenazine mustard (0.1–80 μM), calmidazolium (0.1–80 μM), DT-061 (0.1–80 μM) and OphA (0.3–20 μM) using the K-RasG12V (A) or H-RasG12V (B) nanoclustering-BRET assays in HEK-293 EBNA cells. The A/D plasmid ratio was 4/1. Data represent mean values ± SD, n ≥ 3. The data were fit into log (inhibitor) vs variable response (four parameters) function of Prism (GraphPad) to obtain the dose response curves.
(TIF)

**S4 Fig. Western blotting repeats relating to Fig 4A and 4B.** (**A-D**) Measurement of c-MYC expression after compound treatment in MDA-MB-231 cells from four independent biological repeats. Uncropped Western blot membranes of all biological repeats are presented. After the transfer the membrane was cut between 100 and 70 kDa and the top and bottom blots were

probed with anti-vinculin and anti-c-MYC antibody, respectively. Control indicates 0.1% (v/v) and 0.2% (v/v) DMSO treatments. Marker 1 and marker 2 indicate PageRuler (ThermoScientific) and unstained Precision Plus (Bio-Rad) protein ladders. Note that the anti-c-MYC antibody (#5605, Cell signaling) strongly detected an unknown protein at 25 kDa in MDA-MB-231 cell lysate compared to c-MYC, which was detected in between 55 and 70 kDa.
(TIF)

**S5 Fig. Western blotting repeats relating to Fig 4C and 4D.** MAPK signaling output measurement in MIA PaCa-2 cells upon treatment with control compounds trametinib (1 μM) and AMG-510 (1 μM), single agent treatment with calmidazolium (20 μM) and DT-061 (20 μM) or in combination (DT-061 10 μM + CMZ 10 μM) as well as fluphenazine mustard (20 μM). (**A, B**) Uncropped Western blot membranes of two biological repeats and (**C**) a cropped Western blot membrane from a third biological repeat are presented. The membrane was first probed for ERK and pERK levels simultaneously. After developing the membrane using secondary antibodies, the same membrane was then again probed for GAPDH. X and marker indicate non-related sample and unstained Precision Plus (Bio-Rad) protein ladder, respectively.
(TIF)

**S6 Fig. Proliferation data of Fig 4E.** Comparison of effects of compounds on proliferation of MDA-MB-231 (**A**), MIA PaCa-2 (**B**) and HEK-293 EBNA (**C**) cells grown in 2D monolayers. Cells were treated with compound at concentration ranges of 0.6 μM– 80 μM (fluphenazine and fluphenazine mustard) and 0.003 μM– 40 μM (AMG-510). Data represent mean values ± SD, n = 3. The data were fit to log (inhibitor) vs response–variable slope (four parameters) equation of Prism (GraphPad) software. The actual curve fitting for $DSS_3$ calculation was done on the DSS platform Breeze.
(TIF)

**S1 Data. Raw data of all figures.** Raw data that were used to construct the graphs in **Figs 1–4** are presented in the excel sheet. The header indicates the sample and the rows or columns list values of individual replicates (N). In addition the mean and standard deviation of all figures are given.
(XLSX)

## Acknowledgments

We thank Dr. Karolina Pavic for her critical comments on the manuscript and support with Western blotting, Dr. Rohan Chippalkatti for his support with flow cytometry experiment and data analysis, and Dr. Farah Kouzi for technical support.

## Author Contributions

**Conceptualization:** Daniel Abankwa.

**Data curation:** Ganesh Babu Manoharan, Sunday Okutachi.

**Formal analysis:** Ganesh Babu Manoharan.

**Funding acquisition:** Daniel Abankwa.

**Methodology:** Ganesh Babu Manoharan, Sunday Okutachi.

**Supervision:** Ganesh Babu Manoharan, Daniel Abankwa.

**Visualization:** Ganesh Babu Manoharan.

**Writing – original draft:** Ganesh Babu Manoharan, Daniel Abankwa.

**Writing – review & editing:** Daniel Abankwa.

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
