## [Decision Letter · Decision Letter 0]

20 Sep 2021

PONE-D-21-23656Potential of phenothiazines to synergistically block calmodulin and reactivate PP2A in cancer cellsPLOS ONE

Dear Dr. Abankwa,

Thank you for submitting your manuscript to PLOS ONE. After careful consideration, we feel that it has merit but does not fully meet PLOS ONE’s publication criteria as it currently stands. Therefore, we invite you to submit a revised version of the manuscript that addresses the points raised during the review process.

Please note that the PLOS data availability policy requires authors to make all data underlying the findings described in their manuscript fully available without restriction, with rare exception. Uncropped blot/gel image data must be presented as a supplement or in a public repository according to the preparation guidelines.

We look forward to receiving your revised manuscript.

Kind regards,

Irina V. Balalaeva, PhD

Academic Editor

PLOS ONE

Journal Requirements:

Reviewers' comments:

Reviewer's Responses to Questions

**Comments to the Author**

1. Is the manuscript technically sound, and do the data support the conclusions?

Reviewer #1: Partly

Reviewer #2: Partly

Reviewer #3: Yes

2. Has the statistical analysis been performed appropriately and rigorously? 

Reviewer #1: Yes

Reviewer #2: Yes

Reviewer #3: Yes

3. Have the authors made all data underlying the findings in their manuscript fully available?

Reviewer #1: Yes

Reviewer #2: Yes

Reviewer #3: Yes

4. Is the manuscript presented in an intelligible fashion and written in standard English?

Reviewer #1: Yes

Reviewer #2: Yes

Reviewer #3: Yes

5. Review Comments to the Author

Reviewer #1: The paper is an interesting investigation of the ability of phenothiazine derivatives to synergistically act on two cancer associated targets, calmodulin and the tumor suppressor protein phosphatase. The methodology was well chosen, and the findings are sound. I recommend its publication after some minor revisions, as follows.

1. The Abstract should be more accurate in presenting the compounds investigated in the paper.

2. The structure of the investigated PTZs should be included

3. A Conclusions section should be included.

Reviewer #2: Phenothiazines (PPZs) are widely used as antipsychotic drugs because of their inhibitory activity toward monoamine neurotransmitter receptors. Some of the PPZs bind directly to PP2A holoenzyme and enhance its activity. Recently, PPZ-derived small-molecule activators of PP2A reported exerting anti-cancer effects. Moreover, some of the PPZs have inhibitory activity toward Calmodulin (CaM). CaM and PP2A modulate the Ras pathway. Therefore, targeting both CaM and PP2A simultaneously is a promising strategy for Ras-mutated cancers. However, it remains unclear whether PPZ-derivatives retain both CaM inhibitory and PP2A activatory functions. The authors show here that PPZ-derivative fluphenazine mustard (Flu-M) has anti-CaM activity and claimed that PTZ integrates the synergistic activities of CaM inhibition and PP2A activation within one compound. Their story is potentially interesting, but there is a definite lack of data.

(1) The major problem is that there is no data that Flu-M has PP2A inhibitory effect. The authors need to measure PP2A activity or show that Flu-M treatment promotes the constitution of the PP2A holoenzyme.

(2) Figure 1A: The authors present results from the DRIVE database and argue that the survival of K-Ras mutant and BRAF mutant cells is dependent on the PP2A and CaM genes (P11, L250). However, it is only MDA-MB-231 cells that both PP2A and CaM are in "red: a highly dependent". Authors should state more correctly.

(3) Figure 1B: In relation to Figure 1A, why does calmidazolium treatment induce cell death in A375 to the same extent as in other cells, even though A375 survival is CaM-independent? The same is true for the effects of DT-061.

Minor point.

(1) P3, L47: treatment e.g. of bipolar > treatment of e.g. bipolar

(2) P12, L252: alpha subunit of the PP2A enzyme > alpha isoform of the PP2A enzyme

Reviewer #3: In this manuscript "Potential of phenothiazines to synergistically block calmodulin and reactivate PP2A in cancer cells" the authors present interesting data related to the potential utility of using calmodulin inhibitors and PP2A activators to synergistically modulate downstream pathways and drive cancer cell specific death. Overall, the work is well done and represent an interesting conceptual advance in the field. There are several weaknesses that should be addressed prior to this manuscript being suitable for publication

1. Additional assays should be performed to determine effects and mechanisms of cell death - for example colony formation assays with the drugs alone and in combination would be of interest. Additionally, FACS analysis with Annexin and PI staining would be of interest; western blotting for markers of apoptosis such as cleaved caspase 3 and PARP would also strengthen the conclusions from Figure 1

2. Was the BRET assay in Figure 2B performed with DT-061?

3. In Figure 4 the MIAPACA2 pancreatic cancer cell line was used for the studies - what was the rationale for the selection of this cell line; the cell lines used in Figure 1 should be tested for effects of pERK and tERK and more doses and time points should be used to quantitate drugs effects on MAPK signaling - additionally, the authors should look at pMEK and tMEK expression in these experiments

6. PLOS authors have the option to publish the peer review history of their article (what does this mean?). If published, this will include your full peer review and any attached files.

Reviewer #1: No

Reviewer #2: No

Reviewer #3: No

---

## [Author Response · Author response to Decision Letter 0]

10 Mar 2022

Point by point response to revision on manuscript ID PONE-D-21-23656 entitled “Potential of phenothiazines to synergistically block calmodulin and reactivate PP2A in cancer cells”

Reviewer #1:

The paper is an interesting investigation of the ability of phenothiazine derivatives to synergistically act on two cancer associated targets, calmodulin and the tumor suppressor protein phosphatase. The methodology was well chosen, and the findings are sound. I recommend its publication after some minor revisions, as follows.

(1) The Abstract should be more accurate in presenting the compounds investigated in the paper.

(2) The structure of the investigated PTZs should be included

(3) A Conclusions section should be included.

Responses 1:

We thank the reviewer for their positive comments. We improved the manuscript based on the suggestions of the reviewer: 

1. The abstract was improved and mentions the exact compounds employed in the study.

2. The structures of all investigated compounds are now provided in Scheme 1.

3. A conclusion section has been added.

Reviewer #2:

Phenothiazines (PPZs) are widely used as antipsychotic drugs because of their inhibitory activity toward monoamine neurotransmitter receptors. Some of the PPZs bind directly to PP2A holoenzyme and enhance its activity. Recently, PPZ-derived small-molecule activators of PP2A reported exerting anti-cancer effects. Moreover, some of the PPZs have inhibitory activity toward Calmodulin (CaM). CaM and PP2A modulate the Ras pathway. Therefore, targeting both CaM and PP2A simultaneously is a promising strategy for Ras-mutated cancers. However, it remains unclear whether PPZ-derivatives retain both CaM inhibitory and PP2A activatory functions. The authors show here that PPZ-derivative fluphenazine mustard (Flu-M) has anti-CaM activity and claimed that PTZ integrates the synergistic activities of CaM inhibition and PP2A activation within one compound. Their story is potentially interesting, but there is a definite lack of data.

(1) The major problem is that there is no data that Flu-M has PP2A inhibitory effect. The authors need to measure PP2A activity or show that Flu-M treatment promotes the constitution of the PP2A holoenzyme.

Response 2-1:

We thank the reviewer for the comments, which helped us to improve the manuscript. The activation of PP2A by small molecules such as DT-061 leads to the degradation of c-MYC (Farrington et. al., 2020, JBC), thus by measuring the expression of c-MYC, the PP2A modulatory activity of the compounds could be evaluated. In Fig 4A, the PP2A modulatory activity of fluphenazine and fluphenazine mustard is reported along with DT-061, confirming that all compounds can reduce c-MYC expression levels at certain concentration ranges.

(2) Figure 1A: The authors present results from the DRIVE database and argue that the survival of K-Ras mutant and BRAF mutant cells is dependent on the PP2A and CaM genes (P11, L250). However, it is only MDA-MB-231 cells that both PP2A and CaM are in "red: a highly dependent". Authors should state more correctly.

Response 2-2:

We thank the reviewer for pointing this out and are now stating on P14, L274: ‘Of note, only the MDA-MB-231 breast cancer cell line showed a dependency on KRAS, CALM3 as well as PPP2CA.’

(3) Figure 1B: In relation to Figure 1A, why does calmidazolium treatment induce cell death in A375 to the same extent as in other cells, even though A375 survival is CaM-independent? The same is true for the effects of DT-061.

Response 2-3:

We thank the reviewer for this interesting observation. There are several explanations, one could suggest, however, the major point may be that knockdown of a gene and loss of its protein product (all active sites lost) may have a very different effect than (incompletely) inhibiting a specific site on a protein (one site lost). Moreover, a compound may have other (unknown) related targets. Different (transcriptional) compensatory responses may result.

Moreover, the ATARiS data for the 3 CaM genes, which encode identical proteins, show different dependencies on the 3 CaM genes. This may add a cautionary note to the interpretation of the ATARiS data, as factors such as specific expression level or compensatory responses may have to be factored in. ATARiS data are therefore highly valuable to indicate genetic dependencies, but cannot be equated with inhibitor effects.

Minor point.

(1) P3, L47: treatment e.g. of bipolar > treatment of e.g. bipolar

Corrected as suggested. 

(2) P12, L252: alpha subunit of the PP2A enzyme > alpha isoform of the PP2A enzyme

Corrected as suggested. 

Reviewer #3: In this manuscript "Potential of phenothiazines to synergistically block calmodulin and reactivate PP2A in cancer cells" the authors present interesting data related to the potential utility of using calmodulin inhibitors and PP2A activators to synergistically modulate downstream pathways and drive cancer cell specific death. Overall, the work is well done and represent an interesting conceptual advance in the field. There are several weaknesses that should be addressed prior to this manuscript being suitable for publication.

(1) Additional assays should be performed to determine effects and mechanisms of cell death - for example colony formation assays with the drugs alone and in combination would be of interest. Additionally, FACS analysis with Annexin and PI staining would be of interest; western blotting for markers of apoptosis such as cleaved caspase 3 and PARP would also strengthen the conclusions from Figure 1

Response 3-1:

We thank the reviewer for the interesting and encouraging comments. 

We have carried out suggested flow cytometry and apoptosis marker experiments. 

The effect of inhibitors and combinations on cell apoptosis was analyzed by staining treated cells with annexin V and 7-AAD, and analyze marker intensity per cell using flow cytometry/FACS (Fig 1D; S2 Fig). These data show that the combination of DT-061 (PP2A-activator) and calmidazolium (CaM inhibitor) have a higher activity than the individual compounds alone.

The caspase-3 activity of compounds is now reported using CaspaseGlo assay (Fig 1E), similar to data published in literature (Kauko et al., 2018, Sci Trans Med). 

In addition, the effect of compounds on PARP cleavage using Western blotting is reported (Fig 1F), confirming that the combination induced more PARP cleavage than compounds alone.

(2) Was the BRET assay in Figure 2B performed with DT-061?

Response 3-2:

We have now updated Fig 2B and C showing the effect of DT-061 in the K-Ras/CaM BRET assay. In line with the in vitro experiments in Fig 2A, this compound had no effect.

(3) In Figure 4 the MIAPACA2 pancreatic cancer cell line was used for the studies - what was the rationale for the selection of this cell line; the cell lines used in Figure 1 should be tested for effects of pERK and tERK and more doses and time points should be used to quantitate drugs effects on MAPK signaling - additionally, the authors should look at pMEK and tMEK expression in these experiments.

Response 3-3:

The MiaPaCa cell line was used in order to compare the effect of Flu-M with a clinically approved Ras pathway inhibitor AMG-510. 

We apologize, but within this limited study, with one co-author having left the lab, our experimental capacity was insufficient to perform the requested MAPK-signalling investigation on all cell lines studied. The focus in this last dataset is on a higher activity of Flu-M (Fig 4C,D), and the comparison with the clinical compound AMG-510 in cell culture and an assessment of the selective toxicity against cancer cell lines (Fig 4E). 

In the abstract, we now clearly state that this is a limited study.

Nevertheless, we now report the PP2A modulating activity of Flu and Flu-M by assessing the c-MYC expression level, as this is attributed to the PP2A reactivation by SMAPs, such as DT-061 (Fig 4A,B).

---

## [Decision Letter · Decision Letter 1]

6 Apr 2022

PONE-D-21-23656R1Potential of phenothiazines to synergistically block calmodulin and reactivate PP2A in cancer cellsPLOS ONE

Dear Dr. Abankwa,

Thank you for submitting your manuscript to PLOS ONE. After careful consideration, we feel that it has merit but does not fully meet PLOS ONE’s publication criteria as it currently stands. Therefore, we invite you to submit a revised version of the manuscript that addresses the points raised during the review process.

We look forward to receiving your revised manuscript.

Kind regards,

Irina V. Balalaeva, PhD

Academic Editor

PLOS ONE

Reviewers' comments:

Reviewer's Responses to Questions

**Comments to the Author**

1. If the authors have adequately addressed your comments raised in a previous round of review and you feel that this manuscript is now acceptable for publication, you may indicate that here to bypass the “Comments to the Author” section, enter your conflict of interest statement in the “Confidential to Editor” section, and submit your "Accept" recommendation.

Reviewer #1: All comments have been addressed

Reviewer #2: (No Response)

Reviewer #3: All comments have been addressed

2. Is the manuscript technically sound, and do the data support the conclusions?

Reviewer #1: Yes

Reviewer #2: Partly

Reviewer #3: Yes

3. Has the statistical analysis been performed appropriately and rigorously? 

Reviewer #1: Yes

Reviewer #2: Yes

Reviewer #3: Yes

4. Have the authors made all data underlying the findings in their manuscript fully available?

Reviewer #1: Yes

Reviewer #2: Yes

Reviewer #3: Yes

5. Is the manuscript presented in an intelligible fashion and written in standard English?

Reviewer #1: Yes

Reviewer #2: Yes

Reviewer #3: Yes

6. Review Comments to the Author

Reviewer #1: The authors seriously considered all the comments, and improved the manuscript, which is suitable for publication now.

Reviewer #2: At the initial review, I mentioned the following point.

(1) The major problem is that there is no data that Flu-M has PP2A inhibitory effect. The authors need to measure PP2A activity or show that Flu-M treatment promotes the constitution of the PP2A holoenzyme.

The authors responded to this by measuring the c-Myc protein level as an alternative to measuring PP2A activity. Unfortunately, c-Myc is not an indicator of PP2A activity. PP2A-B56 complex dephosphorylates Ser62 of c-Myc and destabilizing it, on the other hand, PP2A-B55 complex dephosphorylates Thr58 of c-Myc and stabilizing it. Whether Flu-M has a PP2A activating effect is the crux of this paper. Therefore, as I first pointed out, the authors should either measure PP2A activity or show that the formation of the PP2A complex is promoted.

Reviewer #3: The authors have done a great job addressing all of my concerns and comments. The manuscript in my opinion is now suitable for publication and no additional edits are suggested or recommended

7. PLOS authors have the option to publish the peer review history of their article (what does this mean?). If published, this will include your full peer review and any attached files.

Reviewer #1: No

Reviewer #2: No

Reviewer #3: No

---

## [Author Response · Author response to Decision Letter 1]

13 Apr 2022

Point by point response to revision on manuscript ID PONE-D-21-23656R2 entitled “Potential of phenothiazines to synergistically block calmodulin and reactivate PP2A in cancer cells”

Reviewer #2:

At the initial review, I mentioned the following point.

(1) The major problem is that there is no data that Flu-M has PP2A inhibitory effect. The authors need to measure PP2A activity or show that Flu-M treatment promotes the constitution of the PP2A holoenzyme.

The authors responded to this by measuring the c-Myc protein level as an alternative to measuring PP2A activity. Unfortunately, c-Myc is not an indicator of PP2A activity. PP2A-B56 complex dephosphorylates Ser62 of c-Myc and destabilizing it, on the other hand, PP2A-B55 complex dephosphorylates Thr58 of c-Myc and stabilizing it. Whether Flu-M has a PP2A activating effect is the crux of this paper. Therefore, as I first pointed out, the authors should either measure PP2A activity or show that the formation of the PP2A complex is promoted.

Response:

We apologize that the reviewer feels we still have not satisfactorily addressed their concern. 

1) However, as the reviewer will know, production and reconstitution of the PP2A holoenzyme is technically demanding and even PP2A-focused research groups, such as e.g. from Drs. Narla (Farrington et al., 2020; PMID: 31822503) and Westermarck (Kauko et al., 2018; PMID: 30021885) employ the c-MYC destabilization assay as a surrogate to measure PP2A activation.

The former group established in the above publication that the anti-tumor activity of DT-061 depends on the Ser62-dephosphorylation dependent c-MYC degradation. 

Furthermore, the way we introduce c-MYC degradation as surrogate assay, and how we conclude our findings, reflects the limitations of our assay: L. 416- Both Flu and Flu-M decreased the c-MYC level in MDA-MB-231 cells (Fig 4A,B and S4 Fig) comparable to that of DT-061, supporting their PP2A activation. 

2) For the first revision, we unsuccessfully attempted to obtain the PP2A-A/ B56alpha Nano-BRET construct described in Leonard et al., 2020 (PMID: 32315618). However, it was not possible to obtain these constructs from the community, and again such an assay would have had limitations by focusing only on one B subunit.

3) Our major point is that two properties of phenothiazine derivatives, the retention of their CaM inhibitory and PP2A activating properties, could be important to improve them as anti-cancer agents. The CaM inhibition has previously been neglected in the development of PP2A activators. We believe that our admittedly limited study still provides data for thought in ongoing PP2A activator development projects. This conclusion does not only rest on fluphenazine mustard but is likewise supported by fluphenazine and the combination of DT-061 and calmidazolium (Figure 4 A,B).

We have clarified the limitations pointed out by the reviewer with the following statement in the adapted manuscript and hope that the reviewer would accept this, given that in vitro reconstitution studies are out of reach for us:

L. 460- One clear limitation of our study is that we have not directly examined PP2A activation by PTZ, for example by reconstitution of the holoenzyme, but instead employed surrogate measures such as c-MYC degradation.

---

## [Decision Letter · Decision Letter 2]

4 May 2022

Potential of phenothiazines to synergistically block calmodulin and reactivate PP2A in cancer cells

PONE-D-21-23656R2

Dear Dr. Abankwa,

We’re pleased to inform you that your manuscript has been judged scientifically suitable for publication and will be formally accepted for publication once it meets all outstanding technical requirements.

Kind regards,

Irina V. Balalaeva, PhD

Academic Editor

PLOS ONE

Additional Editor Comments (optional):

Reviewers' comments:

Reviewer's Responses to Questions

**Comments to the Author**

1. If the authors have adequately addressed your comments raised in a previous round of review and you feel that this manuscript is now acceptable for publication, you may indicate that here to bypass the “Comments to the Author” section, enter your conflict of interest statement in the “Confidential to Editor” section, and submit your "Accept" recommendation.

Reviewer #2: All comments have been addressed

2. Is the manuscript technically sound, and do the data support the conclusions?

Reviewer #2: Partly

3. Has the statistical analysis been performed appropriately and rigorously? 

Reviewer #2: N/A

4. Have the authors made all data underlying the findings in their manuscript fully available?

Reviewer #2: Yes

5. Is the manuscript presented in an intelligible fashion and written in standard English?

Reviewer #2: Yes

6. Review Comments to the Author

Reviewer #2: (No Response)

7. PLOS authors have the option to publish the peer review history of their article (what does this mean?). If published, this will include your full peer review and any attached files.

Reviewer #2: No

---

## [Editor Report · Acceptance letter]

17 May 2022

PONE-D-21-23656R2 

Potential of phenothiazines to synergistically block calmodulin and reactivate PP2A in cancer cells 

Dear Dr. Abankwa:

I'm pleased to inform you that your manuscript has been deemed suitable for publication in PLOS ONE. Congratulations! Your manuscript is now with our production department. 

Kind regards, 

on behalf of

Dr. Irina V. Balalaeva 

Academic Editor

PLOS ONE